# Sparse Convolution FPGA Accelerator Based on Multi-Bank Hash Selection

**DOI:** 10.3390/mi16010022

**Published:** 2024-12-27

**Authors:** Jia Xu, Han Pu, Dong Wang

**Affiliations:** 1Institute of Information Science, Beijing Jiaotong University, Beijing 100044, China; timxu826@163.com; 2Beijing Key Laboratory of Advanced Information Science and Network Technology, Beijing 100044, China; 3Intel Flex, Beijing 100091, China

**Keywords:** deep convolutional neural network, FPGA, heterogeneous computing, high-level synthesis, cache memory

## Abstract

Reconfigurable processor-based acceleration of deep convolutional neural network (DCNN) algorithms has emerged as a widely adopted technique, with particular attention on sparse neural network acceleration as an active research area. However, many computing devices that claim high computational power still struggle to execute neural network algorithms with optimal efficiency, low latency, and minimal power consumption. Consequently, there remains significant potential for further exploration into improving the efficiency, latency, and power consumption of neural network accelerators across diverse computational scenarios. This paper investigates three key techniques for hardware acceleration of sparse neural networks. The main contributions are as follows: (1) Most neural network inference tasks are typically executed on general-purpose computing devices, which often fail to deliver high energy efficiency and are not well-suited for accelerating sparse convolutional models. In this work, we propose a specialized computational circuit for the convolutional operations of sparse neural networks. This circuit is designed to detect and eliminate the computational effort associated with zero values in the sparse convolutional kernels, thereby enhancing energy efficiency. (2) The data access patterns in convolutional neural networks introduce significant pressure on the high-latency off-chip memory access process. Due to issues such as data discontinuity, the data reading unit often fails to fully exploit the available bandwidth during off-chip read and write operations. In this paper, we analyze bandwidth utilization in the context of convolutional accelerator data handling and propose a strategy to improve off-chip access efficiency. Specifically, we leverage a compiler optimization plugin developed for Vitis HLS, which automatically identifies and optimizes on-chip bandwidth utilization. (3) In coefficient-based accelerators, the synchronous operation of individual computational units can significantly hinder efficiency. Previous approaches have achieved asynchronous convolution by designing separate memory units for each computational unit; however, this method consumes a substantial amount of on-chip memory resources. To address this issue, we propose a shared feature map cache design for asynchronous convolution in the accelerators presented in this paper. This design resolves address access conflicts when multiple computational units concurrently access a set of caches by utilizing a hash-based address indexing algorithm. Moreover, the shared cache architecture reduces data redundancy and conserves on-chip resources. Using the optimized accelerator, we successfully executed ResNet50 inference on an Intel Arria 10 1150GX FPGA, achieving a throughput of 497 GOPS, or an equivalent computational power of 1579 GOPS, with a power consumption of only 22 watts.

## 1. Introduction

In recent years, Artificial Intelligence (AI) has regained tremendous attention and investment due to the emergence of big data and the rapid growth of computing power. Machine Learning (ML) methods have been successfully applied to solve many problems in both the academia and industry fields.

Although the explosive growth of big data applications has driven the development of ML, it has also posed severe challenges to the data processing capability and scalability of the computer systems. More specifically, traditional Von Neumann computers have independent processing and data storage units. The frequent data movement between the processor and the off-chip memory limits the performance and energy efficiency of the system, which is further exacerbated by the rapidly growing amount of data in AI applications. These demands led to the emergence of a category known as’specialized computing’, which focuses on custom solutions tailored specifically for AI applications. In contrast, GPUs provide up to 10 TOPs (Tera Operations Per Second) peak performance, making them an excellent choice for an wide range of neural network (NN)-based applications. Additionally, because of the parallel characteristics computation workload, GPUs has also become the preferred option for neural network acceleration. Beyond CPUs and GPUs, FPGAs are emerging as a promising platform for energy-efficient neural network processing. With hardware designs optimized for neural networks, FPGAs can achieve high levels of parallelism while exploiting the inherent characteristics of neural network computations to eliminate redundant logic. Algorithmic research further suggests that neural network models can be simplified in a manner conducive to hardware implementation, without sacrificing accuracy. As a result, FPGAs have the potential to outperform CPUs and GPUs in terms of energy efficiency.

Many sparse convolution algorithms, particularly those designed for unstructured pruning of networks [1], are inefficient on most parallel computing architectures [2]. This inefficiency arises because the zero values in the parameters are still processed in a vectorized manner, leading to no reduction in computation despite the inherent sparsity. In addition, in most resource-constrained devices, the pruned weights are stored in a compressed data format to save storage. These compressed data need an extra computation load to expand before computing. Both of these characteristics lead to extra computation that is redundant. A well-designed architecture dedicated for sparse convolution can avoid processing zeros as well as extra weight expand phrase, leading to higher energy efficiency and improved performance per watt.

Meanwhile, due to the memory hierarchy and the computational order of the convolution, data reuse plays a significant role in determining execution efficiency. On modern GPUs, the compute speed has outpaced memory speed, with most convolution operations being bottlenecked by memory accesses. As a result, a large portion of execution time is spent on data retrieval. To prevent a substantial gap between actual throughput and theoretical peak TOPs, careful consideration of the convolution execution order and data reuse strategies is essential.

Therefore, the design of dedicated processors as an acceleration platform is increasingly important in response to the rapid growth of AI. Equally crucial is the design of efficient neural network accelerator architectures, which should not be overlooked. This area holds significant potential for exploration, offering substantial commercial value through faster algorithm iteration and seamless cross-platform and cross-device deployment.

Different hardware architectures require distinct optimization strategies. Specifically, for temporal architectures such as Central Processing Units (CPUs) and Graphics Processing Units (GPUs), computational transformations on kernels, such as FFT and Winograd, can reduce the number of multiplications and accumulations, thus improving throughput. In contrast, for spatial architectures used in accelerators, dataflow and pipelining techniques are commonly employed to enhance data reuse from low-cost memories in the memory hierarchy, ultimately reducing energy consumption. Despite the variety of optimization schemes, two critical issues remain: (1) Memory access is often a bottleneck, and efficient memory access design is crucial for maximizing the utilization of computational units, thereby achieving performance closer to the theoretical optimal. (2) Hardware-software co-design enables a better balance between computational load and inference accuracy. A notable example is sparse convolution-based accelerator designs, which demonstrate superior performance [3,4,5].

This paper investigates the design of a hardware accelerator capable of efficiently accelerating sparse convolution algorithms while improving bandwidth efficiency. The main contributions are as follows:

(1) We propose the algorithm and the corresponding hardware structure of a novel sparse convolution scheme. This accelerator design is sparsity-aware, detecting zero values in the weights at any position during the convolution inference operation to reduce unnecessary computations. As a result, the system’s throughput is significantly improved, along with performance per watt.

(2) We propose a shared on-chip cache for storing feature maps, utilizing a hash-based address mapping strategy to support asynchronous convolution. This design reduces on-chip resource occupancy caused by data redundancy and minimizes conflict rates. As a result, higher performance is achieved in the design through efficient space exploration.

(3) We propose a weight encoding method and a hash algorithm tailored for the proposed sparse convolutional neural network accelerator. This ensures the accelerator’s efficiency by preventing on-chip reading conflicts among different computation units. The architecture is deployed on the Intel Arria 1150GX Field Programmable Gate Array (FPGA) and successfully runs ResNet50, achieving a peak computing power of 497 GOP/s and an equivalent performance of 1827 GOP/s.

## 2. Related Work

In this section, we provide a brief review of various general-purpose computing chips and architecture designs, including CPUs, GPUs, system-on-chips (SoCs), standalone dedicated chips, and accelerators designs based on FPGA devices.

### 2.1. General-Purpose Processor

CPUs and GPUs use parallelization techniques such as SIMD or SIMT to execute MACs in parallel. All Arithmetic Logic Units (ALUs) share the same control and memory (register files). On these platforms, Fully Connected (FC) layers and convolutional (Conv) layers are often mapped to matrix multiplications (i.e., kernel computations).

Today’s CPUs—across edge, data center, cloud, and client—include integrated AI optimizations and accelerators that increase AI (Artificial Intelligence) performance and help maximize efficiency and scalability. For example, Intel’s Knights Landing architecture CPU has a superscalar instruction set that supports deep learning; Intel’s Neural Processing Unit (NPU) has a 16-bit floating point (FP16) and int-8/int-4 fixed point arithmetic support and can perform two FP16/int-8/int-4 operations on one single-precision core to speed up deep learning calculations. There are also systems built specifically for DNN processing, such as Intel’s Gaudi DNN server [6]. Intel AMX [7] is a new built-in accelerator that improves the performance of deep-learning training and inference on the CPU and is ideal for workloads like natural-language processing, recommendation systems and image recognition. The Intel AVX-512 [8] accelerator is a set of instructions that can boost performance for vector processing-intensive workloads. Vector processing, an essential part of many advanced computational tasks, performs an arithmetic operation on a large array of integers or floating-point numbers in parallel. RISC-V is a new instruction set that is thriving in the community [9]. Its V Extension and M Extension provide the foundation and convenience of instruction sets for neural network processing. For example, the XuanTie team proposed a custom matrix extension instruction set (MME: Matrix-Multiply Extension) based on the RISC-V instruction set for matrix multiplication operations. In XuanTie C920 [10], the use of a moderate amount of RV64V units also provides good support for AI computing. These built-in accelerators are often less complex to use than discrete accelerators, helping solution designers achieve faster time to value.

Compared to general-purpose CPUs, GPUs have evolved into specialized architectures optimized for parallel processing, featuring SIMD instruction extensions that enable the concurrent execution of multiple tasks. GPUs have processing cores that are far simpler than those of standard high-performance CPUs [11]. This type of architecture works very well in situations where there aren’t many or any branching conditions. In addition, GPU architectures include a memory architecture designed specifically for high-speed data streaming for image processing. Now, with the increasing demand for AI, many extended instruction sets and extended cores have been added to GPUs, such as Nvidia’s addition of Tensor Cores in the Hopper Architecture [12], and gradually added support for multiple quantization bit widths, structured sparsity, and microscaling in the 4-th generation update. With the launch of Intel’s Xe-HPC architecture and dedicated DataCenter GPU (formerly Ponte Vecchio), Xe Matrix Extensions (XMX) built with deep systolic arrays is also inserted to the GPU architecture. Although general-purpose processors can perform neural network inference tasks, they are not specifically optimized for this purpose. For CPUs, the need for extensive functional support leads to increased hardware complexity, and the architecture is not well-suited for large-scale parallel operations. Similarly, while GPUs excel at parallel processing, there remains a mismatch between memory bandwidth and computational power during neural network inference, which limits their execution efficiency.

### 2.2. Dedicated Processors and SoCs

The Neural Processing Unit (NPU) is designed to accelerate computation-intensive operations or kernels within neural network models. These processors are primarily based on loop unrolling techniques or systolic array implementations [13,14,15]. Their computing power basically relies on the multiplication or accumulation unit (Digital signal processor, DSP) in the FPGA to achieve. The Tensor Processing Unit (TPU) [16], with its design concept shown in Figure 1, a TPU consists of 8 Processing Units (PE), and each PE calculates a neuron. TPU can save 97% of CPU instructions and achieve a maximum acceleration of 11.1 times. However, because of the architecture of the systolic array, these designs are unaware of the sparsity in Convolutional Neural Networks (CNNs) and perform poorly on both compute throughput and efficiency. Apple Silicon [17], particularly the M series chip, integrates a CPU, GPU, and a 16-core Neural Engine into a single SoC. Built on a 5-nanometer process, it delivers efficient parallel processing and accelerates machine learning computations, making it suitable for deep learning tasks. Tesla’s Full Self-Driving (FSD) computer [18] features a custom AI chip with a dual neural network accelerator, each chip features. Intel Xeon processors with High Bandwidth Memory (HBM) [19] offer high memory bandwidth and reduced latency, essential for deep learning workloads. The combination of Xeon CPUs and HBM provides fast access to large datasets and efficient parallel processing, making them ideal for data centers and high-performance computing environments. The NVIDIA Jetson Orin [20] platform features an Ampere architecture GPU, Arm Cortex-A78AE CPU, and a deep learning accelerator, delivering up to 200 trillion operations per second. It supports various AI frameworks, making it suitable for edge devices and applications requiring real-time AI processing, such as robotics and autonomous vehicles. However these dedicated processors are not specifically designed for unstructured sparse neural networks, or their parallel processing architecture cannot achieve good energy efficiency gains in performing unstructured pruning neural networks.

### 2.3. Overview of FPGAs-Based Soft Accelerators

FPGA based target detection accelerator designs can be roughly divided into three categories: (1) The first category is mainly the most common convolution hardware implementation. These algorithms are basically based on loop unrolling technology or systolic array implementation [13,14,15]. Their computing power basically relies on the multiplication or accumulation unit (Digital signal processor, DSP) in the FPGA to achieve.

(2) The second type of processor focuses on converting the multiplication and accumulation calculations of DSP blocks into operations with lower bits (such as converting floating-point numbers to fixed-point numbers) or binary operations, such as hardware-optimized bit shifts or XOR operations, and can use the logic resources (regslice) in the FPGA to implement such operations at low cost [22,23]. Since implementing such operations will consume less logic resources and save the trouble of DSP wiring, this method is usually no longer bottlenecked by the number of DSPs, thus achieving higher performance. For example, in [24], the authors implemented the binary part of the backbone network in the YOLOv2 network and deployed the entire network on a Xilinx Zynq Ultrascale FPGA, achieving an inference performance of 40.81 FPS. Similarly, in another work [23] the authors implemented a mixed-precision YOLOv2 network structure and achieved performance comparable to Lightweight-YOLOv2 [25]. LUTNet [26] greatly increases area efficiency by replacing resource-intensive multipliers with lightweight XNOR gates. In the past two years, with the rise of binary networks, simple operators have become very friendly to FPGAs because their main calculations are bit operations and the memory requirements are greatly reduced. FracBNN [27] retains the main advantages of traditional uses of Binary Neural Networks (BNNs), in which all convolutional layers are calculated with pure binary MAC operations (BMACs). StereoEngine [28] BNNs to learn discriminative binary descriptors and realize the application of binary networks. However, a significant disadvantage of using low-bandwidth quantized networks is that the representation ability of low-bandwidth parameters is poor, which will cause serious network accuracy loss and will have a relatively large limitation in scenarios with high model recognition accuracy.

(3) The third type of processor focuses on the sparse neural network. Previous research, such as dense-sparse-dense (DSD) [29], has demonstrated that a significant portion of neural network connections can be pruned to zero with minimal or no accuracy loss. Various computing architectures have been proposed to leverage this sparsity. For instance, EIE [30] is designed to accelerate computations in NN models with sparse weight matrices and sparse feature maps, respectively. In this field, the key design focus is on the control logic and redundant computing reduced to achieve higher performance, as well as performance per watt. Several algorithmic approaches have been explored to design NN models in a hardware-efficient manner, such as utilizing block sparsity. Techniques to manage irregular memory access and unbalanced workloads in sparse NNs have also been developed. For example, Cambricon-X [31] and Cambricon-S [32] address memory access irregularities in sparse NNs through a collaborative software/hardware approach. Still, a group of soft accelerators based on FPGA is exploited. TCAS1’22 [33] propose a hardware/power-efficient and highly flexible architecture for supporting both unstructured and structured sparse CNNs, featuring an efficient weight reordering algorithm, an adaptive Hybrid Parallel (HP) on-chip dataflow for weight reuse, and an off-chip partial fusion scheme. VLSI’21 [4] propose a sparse-wise dataflow that skips cycles of processing multiply-and-accumulates (MACs) with zero weights and utilizes data statistics to minimize energy through zeros gating, leading to low bandwidth requirements and high data sharing dedicated to structured pruning. FCCM’20 [34] presents a dataflow designed architecture to optimize homomorphic encryption for sparse convolutional neural networks data reuse, coupled with an efficient scheduling policy that minimizes on-chip SRAM access conflicts. These accelerators either do not utilize streaming, or are designed for standard convolution and low-bit convolution, respectively. The third category targets sparse convolution, but they still fail to fully exploit the potential performance gains offered by sparse convolution and low-bit quantization. In contrast, we explore the deeper relationship between sparsity and low-bit characteristics, while also optimizing work balance and cache design to achieve superior performance.

## 3. Design of Sparse Convolution Calculation Acceleration Circuit on FPGA

This section reviews the current use of neural network computing devices and the main challenges they face. It then introduces the design of a neural network accelerator proposed in this paper to address these issues. The paper also provides a detailed description of the overall architecture, including data flow, control logic, and read/write strategies. Finally, the accelerator is deployed on FPGA, and the optimization results, along with design bottlenecks, are analyzed through experimental evaluation and modeling.

### 3.1. Review of Unstructured Pruning Approach

When discussing the cost of deploying neural networks, parameter count and FLOPS (floating-point operations per second) are key metrics. Networks often have billions of weights, which are typically associated with high performance. Previously, in the field of model compression, neural network model pruning often used the pruning of model parameters as the only goal, because in sparse convolution calculations, reducing model parameters means setting some values in the model weights to zero, and zero values do not participate in the calculation [23], which can be regarded as reducing the amount of calculation or improving the equivalent computing power.

Unstructured pruning generally operates at the granularity of individual neurons, in contrast to structured pruning [35], which is based on filters, channels, or even entire layers. Unstructured pruning prunes parameters directly, which offers several advantages. It is straightforward, as setting a weight to zero within the parameter tensors effectively prunes a connection. Deep learning frameworks like PyTorch provide easy access to all network parameters, simplifying implementation. The primary benefit of pruning connections is that they are the smallest and most fundamental elements of networks, allowing for extensive pruning without affecting performance. This fine granularity enables the pruning of subtle patterns, even within convolution kernels. This weight pruning is unconstrained and the most precise method. This difference in granularity results in irregular pruning patterns for unstructured pruning [36], whereas structured pruning yields more regular patterns. For general-purpose accelerator hardware such as CPUs and GPUs, structured pruning is more effective in deployment due to its regular sparse granularity and unstructured pruning may not provide performance improvement. However, for specialized neural network processors, unstructured pruning is more attractive as it can better reduce the memory footprint and computational workload of DNN models. As illustrated in Figure 2, a pruned weight can be skipped right in the convolution operation, in a dedicated designed architecture. Recent foundational research on unstructured pruning has focused on effectively training and identifying “winning tickets” [37,38], which are subnetworks within dense neural networks that can match the original network’s test accuracy after the same number of training iterations.

### 3.2. Top-Level Design of the Sparse Neural Network Accelerator

This section provides a detailed introduction to the hardware architecture design of the sparse neural network accelerator, specifically the multiple cores of the accelerator deployed on the FPGA in this study. The design of the kernel and the data flow are key elements discussed in this chapter.Therefore, the following elaborates on the access unit, the computing unit, the shared memory unit, and introduces the design work of the internal logic circuit and the storage circuit in each unit. As shown in Figure 3, the accelerator designed in this article contains a task scheduler, a feature map memory (Feature Buffer in the figure), a data loading unit (Data Load Unit, DLU, memRead in the figure), a data storage unit (Data Store Unit, DSU, memWrite in the figure), multiple SCUs, and interfaces connecting external Double Data Rate (DDR) storage and PCIe bus. Among them, DLU is used to read feature maps and weight files, and then distribute (Map) feature maps and weight files to feature map memory and computing units respectively; DSU is responsible for collecting (Reduce) the convolution output feature maps completed in the computing unit, storing them back to the main memory and completing the task signal; SCU is composed of Accum, Post, and Mult in the figure, and the task flows composed of them are pairing multiplication, and accumulation operations of feature maps and weights. The task scheduler is used to start and perform synchronization operations between computing units and feature map memory and is responsible for receiving instructions and sending running status and feature map data. The task scheduler collects task execution status between modules and chooses to start data reading and data writing. It is also responsible for the start and stop status of tasks and the distribution and balance of convolution operations. At the same time, the accelerator contains an optional Maximum Pooling Unit (MPU), which can choose to complete pooling operations in FPGA or CPU according to different task strategies. Under the above structural functions, we also design the software and hardware systems for online and offline computing loads, so that the loads between different computing units in inference are balanced and a set of consistent feature map memories are shared. This effectively avoids a lot of waiting and synchronization caused by maintaining data consistency, which is conducive to improving the computing efficiency of the overall accelerator.

It is worth noting that when designing the OpenCL code, this article configures the DLU, DSU, and MPU as single-threaded acceleration kernels, passing parameters through the OpenCL host. The SCU responsible for convolution calculations is configured as an automatically running acceleration kernel, passing parameters through the OpenCL inter-kernel channel. The SCU codes for multiple asynchronous parallel calculations are the same, and the instruction __attribute__((num_compute_units($N_{scu}$))) is used to instruct the compiler to generate multiple identical circuits during the synthesis stage. The parallelism in each SCU is achieved by loop unrolling using the #*pragma* unroll instruction.

### 3.3. Task Scheduling Solution

Since the order of grouped convolutions is rearranged, the sequence of feature maps is disrupted when storing the output feature maps. To address this, we propose a sequential recording mechanism that ensures the output feature maps are written to their correct locations. In the memRead kernel, the channel number (channel id) of the current execution is directly sent to the memWrite kernel. In the memWrite kernel, the channel ID is received and added to the offset of the address where the output feature map is written, so that the output feature map position can be located.

**Task-Level Load Balancing.** After pruning, the workload of each channel is unbalanced. The following figure shows the number of non-zero weights (workload) of each channel in the 12-th layer of a pruned VGG16 model:

For some areas, such as when channel < 64 or channel > 400, the workloads of adjacent channels differ greatly, which reduces the efficiency of the convolution circuit when performing synchronous convolution. To solve this problem, we propose a load-balancing algorithm. First, sort the channels in descending order according to the workload size. Then, divide the adjacent channels into a group (the group size is the number of parallel convolution units, that is, the parallelism in the channel direction). Then, during the convolution operation, the convolution calculations of the two adjacent groups are loaded at the same time. Meanwhile, the channel calculation order of the latter group in the two groups is reversed (so that the workload is reversed to the other group). Then, the two calculation workloads of the corresponding channels are merged so that the calculation loads of each channel can be made closer. The effect is shown in Figure 4:

It can be clearly seen from Figure 5 that after using the proposed channel load balancing scheme, each layer basically shows a step-by-step decline, which means that after each data distribution, the load of each computing unit is basically the same. It can be seen from the bottom of Figure 6 that several mainstream backbone networks have achieved good optimization results.

**Memory Access Control within the Computing Unit.** For an accelerator with $N_{SCU}$ computing units, each SCU calculates an output channel at the same time. When calculating a channel, it obtains data from all groups in a certain order, but the same group can only read data for one SCU at the same time. In principle, we require the number of groups to be twice the number of SCUs to reduce conflicts. Use static scheduling to reduce congestion between banks. And write this scheduling method into the quantization table (quant_table). Algorithm 1 shows the specific scheduling scheme.
**Algorithm 1:** Packet Shared Buffer Static Scheduling**Input**: The number of banks M; the number of SCUs N; Time[M][N];**Output**: $\{P_{1},P_{2},\ldots ,P_{n}\}=P$ denote the bank that each SCU has accessed;   1: $\{S_{1},S_{2},\ldots ,S_{n}\}\in S$ as the current end time of each SCU;   2: Set A is the set of available banks;   3: R = $R_{1},R_{2},\ldots ,R_{n}$ is the access order (forward and reverse);   4: fori=1:Ndo   5:    bk=random_choice(A);   6:    $P_{i}.push\_back\left(bk\right);$   7:    $S_{i}=Time\left[bk,i\right];$   8:    $A.remove(S_{i});$   9: end 10: $While\;for\;each\;P_{i}\;in\;P\;:\;P_{i}<M$ 11:    i=argmin(S) 12:    A.insert(i); 13:    forjinA 14:       $if\;j\;not\;inP_{i}$ 15:          A.remove(j); 16:          $P_{i}.push_{b}ack\left(j\right);$ 17:          $S_{i}+=Time\left[j,i\right];$ 18:       end 19:    end 20: end

**Channel Balancing.** In terms of scheduling different convolution kernels, the proposed parallel computing mode of pre-fetch window parallel partitioning is illustrated as in Figure 7:

For larger feature maps, shown in Figure 7 take 4 feature map blocks $N_{in}\times N_{y}\times C$ of size $N_{in}\times N_{y}\left(N_{in}=N_{y}\right)$ on the H×W plane. Put feature map blocks 1, 2, 3, and 4 into Group1, Group2, Group3, and Group4 in Figure 7 respectively, and the four groups execute in parallel. Each Group has $N_{y}Y$ SMs executing in parallel. Each SCU operates on 1 kernel and the feature map of the current group. All SCUs reuse the same set of feature maps; there are $N_{y}$ SCUs calculating in parallel. The SCUs in the four groups share the same kernel. Inter-Kernel Channel Balance rearranges the parallel calculated Kernels in each Group so that the calculation time between different SCUs in each group is similar. In each SCU, there are $N_{y}$ SCUs calculated in parallel. Each SCU performs multiplication and accumulation (MAC) operations on the 1×1 strips 1×1×C of each kernel on the SxR plane. $N_{y}$ SCUs perform convolution operations on one strip, and $N_{y}$ strips from different kernels are operated in one Group at the same time. If no reordering is performed, the convolution operations are performed in the order of 1, 2, 3, 4, 5, 6, 7, 8, 9. Different kernels are run in different SMs, that is, different kernels No. 1 are run at the same time, and so on. Because the non-zero values are not the same, the time to complete the convolution operation is also different. Rearranging each kernel can reduce the waiting time.

### 3.4. Feature Map Address Mapping

Mapping feature maps into groups and organizing them into segments during convolution is the method used by the accelerator to schedule tasks for the computing units during operation as illustrated in Figure 8. We use on-chip SRAM to store feature maps. Because the stored feature maps are non-sparse, each SCU accesses the feature maps according to the location of non-zero weight values (Weights). Considering that SRAM has only one write port and one read port, the SRAM in the same group can only read feature map data at one address at the same time, that is, the same clock can only serve one SCU. To this end, we divide the feature map memory into multiple groups, that is, multiple banks, and there is no congestion problem when different SCUs access different banks. It is necessary to consider choosing a simple and efficient method to divide the feature map into different groups so that the number of memory accesses to obtain the feature map corresponding to the weight on each convolution kernel is as equal as possible. When storing the feature map, we assume that the feature map corresponding to the weight is randomly and equally distributed, and using direct mapping can reduce complexity. Therefore, using the Least Significant Bit (LSB) or the Most Significant Bit (MSB) is able to map the feature map to the memory of different groups with high accuracy. Experiments show that we use LSB as the code to distinguish banks to obtain higher performance.

#### 3.4.1. Computing Unit Synchronization Scheme

An important job of the task scheduler is to control the synchronization of convolution. In sparse convolution operations, feature maps are often not sparse (or even if sparse representation is used, decoding is required before calculation to complete the calculation). Therefore, the multiplication and accumulation operations of different convolution kernels on the same part of the feature map at the same time are the most commonly used parallelism designs when designing sparse convolution multiplication. In this process, the different sparsity rates in different convolution kernels caused by sparse perception cause synchronization problems in data reading. For example, the difference in reading feature maps brings additional pressure to the feature map buffer. In previous work, people designed independent memories for each operation unit under this parallelism [39] to solve the access conflict problem in asynchronous convolution, which causes multiple copies of data to be stored in the on-chip cache. This data redundancy sacrifices a lot of storage resources. Partial synchronous convolution uses synchronization signals to control the start and stop of convolution units, which causes efficiency reduction due to the waiting synchronization of convolution units in data imbalance. Therefore, we use synchronous convolution with a sorting function to solve the above problems, which is illustrated in Figure 9.

After selecting the operation order of the output channel, we use memRead to coordinate the synchronization signal. After all SCUs are ready, the synchronization signal is sent, and all convolution units start to operate synchronously. After each SCU completes the operation task of the current output channel, it sends the Ready signal to MemRead. When memRead receives the signals from all SCUs, it sends a synchronous start signal. At this time, all SCUs start at the same time. This is to cooperate with static scheduling (we discuss it in detail in Section 3.5.1). In static scheduling, we ensure that the threads are started synchronously, otherwise, the delay caused by external factors may make the scheduling strategy invalid.

#### 3.4.2. Parallel to Serial Conversion

We use 8-bit precision feature maps and weights for convolution operations in the accelerator, but in order to ensure the accuracy of the data, the accumulated data are temporarily stored as 24 bits. When doing the rounding operation, we reuse the rounding device through the parallel-to-serial operation to save on-chip logic resources. Specifically, the parallelism of convPost is $N_{y}\times N_{in}$, that is, every time a set of convolutions is completed, it means that a matrix of size $N_{y}\times N_{in}$ is generated. When doing convolution operations, not every clock has so much data output. For a kernel of size K×K×C, the number of clocks required to complete a set of convolutions, that is, to generate a matrix of size $N_{y}\times N_{in}$, is $K\times K\times C\times P_{\left(params,i\right)}$. ConvMult is an operation that rounds the results of multiple points and accumulates biases, so it does not work all the time. Therefore, we can transfer the generated matrix to convMult through the parallel-to-serial method to achieve the effect of time-sharing multiplexing of the rounding device, thereby using fewer resources. The mechanism is shown in Figure 10.

### 3.5. Cache Design

To address the issues observed above, we design a cache-sharing method using hash mapping. On the computation side, the required feature map is indexed using the logical address, while on the storage side, a translation unit maps the logical address to the corresponding physical address. For the centralized feature map memory, we use multiple groups to store all feature maps, and then distribute all feature maps evenly in each group according to the established hash algorithm. Finally, the scheduler uniformly accesses multiple groups.

#### 3.5.1. Continuous Caching and Data Reuse (Prefetch Window Design)

As mentioned in Section 1, data loading and reuse are very important, so we carefully designed an efficient data loading unit. The data loading unit is designed using strategies such as sliding windows and row registers, illustrated as the flowchart in Figure 11. In the following, we describe the design of the storage circuit from the perspective of loading and reading in Figure 12. We know that the convolution kernel slides on the feature map to perform an accumulation operation, so in the same row, two adjacent convolution operations generally slide to the right. Let’s take a convolution kernel with a size of 3×3 and a stride of 1 as an example: the feature map corresponding to the multiplication of the first point of the second convolution kernel is consistent with the feature map corresponding to the second point in the first convolution, and the sliding window is based on this idea. The data read once can make the reusable data in multiple convolutions not have to be read repeatedly, thereby improving efficiency. On the other hand, when a convolution kernel slides horizontally, any value in it is multiplied with a row of continuous feature maps in the same row, and this part can be operated at the same time. Therefore, the union of a row of feature maps corresponding to each value on the convolution kernel and the number required in the sliding window is still a continuous row. We retrieve this row of continuous feature maps from outside the chip and store them in the feature map memory. Figure 13 illustrates the design of the row memory.

We design a row storage strategy for data reuse and continuous reading, which bring new challenges to addressing, so we establish an algorithm to introduce the addressing method. We mention later that the weights file represents the position of non-zero values. This is used as the position offset for the convolution operation. In 2D conv, the convolution kernel slides zig-zag from left to right and from top to bottom in the x and y directions, respectively, and is recorded as $step_{x}$, $step_{y}$. According to the weight file, the indexable position is [$step_{x}+x$, $step_{y}+y$, depth]. For the structure with $W_{f}$ group buffers, we use the last n bits of the depth encoding as the bank selection, where the selection of n is determined by the hardware design, which is $n=log_{2}W_{f}$. In the actual design, because depth is close to MSB, we use the following method to select the bank. At the same time, in order to achieve fast address conversion, we use a fast Algorithm 2 to optimize the conversion efficiency:
**Algorithm 2:** Quick address selection**Input**: Hardware register group number B, memory access effective      address bit N, input address $Add_{O}$**Output**: Buffer Bank number **SEL**, intra-bank address $Add_{n}$   1: Create an address queue **Que**   2: Add the input address $Add_{o}$ to the queue **Que**;   3: If  !**Que**.empty():   4:    Add = **Que**.pop();   5:    Determine that the Add valid bit is s bit;   6:    Swap the high s−1,⌊s/2⌋ with the low ⌊s/2⌋,0.   7: end

### 3.6. Hash Algorithm

Bank conflicts can significantly reduce the bandwidth of interleaved multi-bank memories, while stalls caused by these conflicts can increase latency in caches or scheduling units. Both issues stem from the same problem: pairs of feature maps, which should be mapped to different indices, are instead mapped to the same index when accessed simultaneously. A suitable hash function can avoid conflicts in each of these cases by mapping the most frequently occurring patterns without conflicts. This section explores different hash functions that meet the criteria and analyzes their performance.

To improve the fit and feasibility of implementing the hash algorithm, we conduct several experiments by selecting and evaluating various suitable hash algorithms. The selected hashing algorithms employ various techniques to efficiently map data to hash values, each offering a unique approach to handling collisions and ensuring an optimal distribution of hash values. XOR-based hashing [40] is a technique that uses the bitwise XOR (exclusive OR) operation to combine and mix input data, producing a hash value that is typically well-distributed and resistant to patterns in the input. Linear probing [41] is a collision resolution method in hash tables where, upon encountering a collision, the algorithm searches for the next available slot in a sequential manner, which can lead to clustering but is simple to implement. The one-at-a-time hash [42] (a.k.a Jenkins Hash), developed by Bob Jenkins, processes input data one byte at a time using a series of bitwise operations such as shifts and XORs, providing a good distribution with minimal computational overhead. Perfect hashing [43] is a technique that creates a collision-free hash function for a known set of keys, ensuring each key maps to a unique hash value, which is particularly useful for static datasets. The mid-square [44] method involves squaring the key and extracting the middle portion of the result as the hash value, offering a simple yet effective way to achieve a good distribution, especially for numeric keys. The folding method divides the key into equal-sized parts and combines them, typically by addition, to produce the hash value, which helps in distributing hash values more uniformly. Modulo arithmetic, a common technique, involves dividing the key by a prime number and using the remainder as the hash value, which helps in reducing collisions and achieving a more uniform distribution. Lastly, the revert LSB [45] technique manipulates the least significant bit of the key to enhance data mixing and distribution, often used in conjunction with other hashing methods to improve overall performance. Each of these hashing techniques offers distinct advantages and is chosen based on the specific requirements of the application, such as dataset size, collision handling, and performance needs.

We apply the following algorithms to ResNet50 using real data, deploy them on the accelerator, and compare the experimental results. The performance data are reported in Table 1:

We aim to use these methods as a bank selector to avoid conflict. So we count the conflicting query divided by the total queries denoted as Conflict Rate to evaluate the main performance of a hash algorithm. The logical complexity and memory usage are also. We find that in the actual design process, our design requirements were very clear, that is, on the one hand, the algorithm latency should be as low and fixed as possible, and on the other hand, it should be as simple as possible to save resources in hardware deployment, because the resource overhead of implementing the algorithm is proportional to the size of $N_{scu}$. Therefore, we finally choose Revert LSB Hash as the actual deployment method and use it as the basic algorithm for performance testing.

### 3.7. Data Encoding

In designing the instruction architecture, we consider the data-intensive nature of CNN operations. To optimize performance, data and instructions are transmitted separately from the outset. During convolution, instructions, weights, and feature map data are sent separately to the FPGA core. The following section introduces the encoding method for instructions and data for performing convolution.

#### 3.7.1. Weight Data Encoding

The weight file is divided by the number of hardware buffer groups and stores the x, y, and C positions of each kernel non-zero element in the order of the group, and encodes the position information into the weight information in the form of [x, y, C] with a bit width of 16 bits. Considering that the sizes of x, y, and C in different convolutional layers are different, the x, y, and C data in each layer are represented by different bits, and the number of bits of each variable is reflected in idx_offset.

As shown in Figure 14, the weights are divided into BANK_NUM groups (banks) according to the encoding method. Each group has $\left\{M_{1},M_{2},\ldots M_{n}\right\}\in W_{f}$ non-zero element weights. To match the hardware, we group the non-zero elements WT_VEC_SIZE and align them with WT_VEC_SIZE. If they are insufficient, fill them with 0. Then the size of each kernel is:(1)$$\sum i=1^{n}\left(\frac{M_{i}}{WT_{VEC_{SIZE}}}\right)$$

#### 3.7.2. Weight Data Decoding

There are two ways to store the location of non-zero values. One is to directly record the index, and the other is to record a sparsity map and calibrate the non-zero element position information through the 1/0 position. The weight value is quantized using 8 or 4 bits. Assuming that the index is encoded using M bits and the data is encoded using N bits, it is necessary to analyze the value of M based on the sparsity rate to find the highest efficiency.

Assuming the total amount of encoded data is *E* and the sparsity is Sp, the total amount of data after encoding is:(2)$$E=\left(\begin{array}{c} N+M\left(1-Sp\right) \end{array}\right)$$

When using low-bit quantization, we use run-length encoding, as shown in Figure 15 and allocate a complete quantization table for each kernel (or output channel), with a size of $W_{f}*3+3$ numbers.

The quantization table serves as a counting mark for the start-stop and working mode of the computing unit to determine the working status of each computing unit at that time and assist the scheduling unit in scheduling operations. Because a computing unit is responsible for the convolution calculation of each channel at the same time, we design a quantization table for each channel, shown in Figure 16 which records the total number of non-zero elements in each channel as the total counter of the convolution task, and the number of groups required to be requested for each channel can help the convolution computing unit identify the groups that do not need to be requested, thereby reducing the time wasted by the computing unit repeatedly initiating and waiting for requests to the scheduler of the feature map memory.

At the same time, the computing unit does not need to distinguish the groups to which the request belongs but is determined by the order of the weight values. Therefore, recording the number of weight values and group numbers required to be requested by each group can enable the computing unit to lock the correct group to avoid conflicts.

## 4. Experiment Results

### 4.1. Experiment Settings

The pruning schemes used for model compression in this paper, including the structured pruning scheme proposed earlier, are implemented using PyTorch [48]. We have adopted the pruning algorithm of GenExp [1]. The benchmark model is ResNet50, and the final performance drop in terms of Top-1 accuracy is 1% when compared with the source full-precision model. The hardware configuration includes an NVIDIA GTX 3090 GPU, an Intel(R) Xeon(R) CPU E5-2678 v3 operating at 2.50 GHz, and 64 GB of memory. For development tools, this project is built on Intel(R) FPGA SDK for OpenCL(TM), Version 20.1.0. The compilation tool used is Intel(R) FPGA SDK for OpenCL(TM) Kernel Compiler. The kernel code is all developed in accordance with the OpenCL™ 1.1 standard.

The sparse convolutional neural network accelerator designed in this paper is implemented on a PCI-Express card of DE5-Net, which is equipped with an Intel Arria-10 GX1150 FPGA. The FPGA device has 427,200 ALMs (1150K LEs), 1518 DSP blocks, and 2560 M20K on-chip memory modules. There are two 4 GB DDR3 SDRAMs on the board, operating at 1600 MHz, which can provide a maximum (operating at full frequency) external memory bandwidth of 25.6 GB/s. The FPGA board is installed on a Z490 chipset equipped with an Intel(R) Core(TM) i9-9900K CPU @ 3.60 GHz and 64 GB DDR4 4000 MHz memory. The OpenCL kernel code is compiled using the Intel FPGA OpenCL SDK v20.1.

### 4.2. Accelerator Performance Modeling

In this section, we employ the Roofline Model, a widely used performance analysis framework, to evaluate the performance and efficiency of the accelerator. Additionally, we construct a resource model based on the accelerator architecture to analyze the utilization of on-chip resources.

#### 4.2.1. Roofline Model

As mentioned in the Roofline Model [49], the maximum theoretical performance of a given hardware processor consists of two aspects: on-chip computing resources and external memory bandwidth. The Roofline Model is a visual performance model that can estimate the performance of a single-core, multi-core, or accelerator processor architecture for a specific calculation or application.

In some network layers, the computational intensity of the algorithm is very low, and the storage bandwidth becomes a bottleneck. At this time, the external memory access bandwidth cannot provide sufficient data for the on-chip computing resources, which results in the idleness of local on-chip computing resources and the situation of waiting for the data required for calculation. Finally, the overall computing performance (i.e., the vertical axis, throughput) cannot meet the theoretical peak performance. When the computing intensity continues to increase, the performance that the processor can obtain for the maximum calculation also increases until it reaches the limit of computing resources; that is, at this time, the time required for the on-chip calculation of the data is very long, and the external memory access bandwidth is sufficient to obtain the data in this time, so the on-chip computing resources do not need to wait for the data to be transmitted, and the load of the entire system can be guaranteed. As shown in Equation (Equation 2) mentioned later, in the structured pruned CNN model, the computational intensity of the *j*-th layer $I_{j}$ decreases with the increase of the pruning rate. Therefore, in theory, there is a maximum value of the sparsity rate $PR_{j}$ in this layer, which can ensure that the *j*-th layer can be in the restricted area of computing resources on the surrounding models when the pruning rate is lower than that, thereby achieving a higher theoretical throughput rate. In other words, when the sparsity rate of the neural network model is at the intersection of the storage range and computing resources of the Roofline Model, that is, the upper left corner, the system can achieve a balance between accessing data and computing data.

Since the computation in this paper is affected by the sparsity rate, the sparsity rate $PR_{j}$ at the theoretical maximum efficiency for the *j*-th convolutional layer can be calculated by the following equation: (3)$${PR}_{j}=1-\frac{I_{j}\times N_{in}\times H_{f_{j}}}{N_{in}\times O_{acc_{j}}-2\times I_{j}\times G_{x_{j}}\times G_{y_{j}}\times P_{j}}$$

We evaluate the system’s efficiency using this model and propose an optimization strategy based on the findings in subsequent experiments.For the two axes of the model, they correspond to two optimization directions. One is to reduce the amount of computation and the other is to reduce the data access. For the ResNet50 network, in the purposed design, the pruning strategy can be optimized from the two dimensions of computation, and data respectively, but because the feature map is dense. Therefore, we are biased to think that the main contribution of pruning is to reduce the computation. On the other direction, i.e., data access, a large part of data access relies on data reuse, and as mentioned in Section 3.5.1, the efficiency of data reuse is related to the size of the convolutional kernel. Especially in deep networks, the data reuse rate decreases with the size of the convolutional kernel K and the deepening of the number of channels. In this paper, based on experiments with ResNet50, we choose the minimum value of WT-Buffer depth $D_{w}$ of 1024 when configuring the shared feature map memory.

#### 4.2.2. Performance Modeling

**Theoretical Arithmetic Model.** The theoretical computation time $T_{l}$ for the convolution of the first *l* layer can be calculated by Equation (Equation 4):(4)$$T_{l}=\frac{P_{ops_{l}}}{N_{in}\times N_{y}\times N_{scu}}\times \frac{1}{Freq}$$
where freq denotes the clock frequency of the convolutional computation unit, while $P_{ops_{l}}$ denotes the amount of sparse post-computation for the convolution of the *l*th layer.

For a specific neural network network model, the overall performance on the architecture of this paper QPS (Query Per Second) is defined by Equation (Equation 5):(5)$$QPS=\frac{1}{\sum T_{l}}$$

**Bandwidth Model.** For the convolution of the *l*-th layer, we use prefetching from the off-chip feature map. In the $C_{l}\times R_{l}\times N_{l}$ three dimensions of the feature map, it takes a total of $G_{l}^{W}$, $G_{l}^{H}$, and $G_{l}^{M}$ prefetches to complete the full calculation, $G_{l}^{W}$, $G_{l}^{H}$, and $G_{l}^{M}$ obtained from Equation (Equation 6).
(6)$$\begin{array}{c} G_{l}^{H}=Ceil\left(\frac{R^{\prime }}{N_{y}}\right)\\ G_{l}^{M}=Ceil\left(\frac{M}{N_{scu}}\right)\\ G_{l}^{W}=Ceil\left(\frac{C^{\prime }}{N_{in}}\right) \end{array}$$

Therefore, for the entire network model, the input feature map size (Byte) to be read from the off-chip DDR is shown in Equation (Equation 7) as follows.
(7)$$H_{f}=\sum_{l}\left(G_{l}^{M}\times \left(N_{in}+K\right)\times s\times \left(N_{y}+K\right)\times s\right)$$

The size of the weight value (Byte) to be read from the off-chip DDR is shown in Equation (Equation 8): (8)$$H_{w}=\sum_{l}\left(P_{params,l}\times K\times K\times C\times N\right)$$

The bias size (Byte) to be read from the off-chip DDR is shown in Equation (Equation 9): (9)$$H_{b}=\sum_{l}N$$

In summary, the average memory bandwidth (Byte/s) is shown in Equation (Equation 10):(10)$$H_{total}=\left(H_{w}+H_{b}+H_{f}\right)\times QPS$$

#### 4.2.3. On-Chip Resource Utilization Model

After compiler optimization, the generated Register-Transfer Level (RTL) codes should align with the estimated usage of logic resources, DSP computational resources, and on-chip RAM specified by the logic resource model. However, adaptive optimizations performed during runtime deployment for different FPGA chips may introduce resource allocation biases.

**RAM Resources.** In the proposed architecture, the global memory serves as the primary cache for loading data into Load-Store Units (LSUs), while the shared memory is implemented using RAM blocks as storage units. Consequently, the utilization of RAM resources is divided into three main components: the on-chip cache, weighted memory, and the global memory for LSUs.

**On-Chip Cache.** In Intel FPGAs, there is typically dedicated memory SRAM on the chip and the depth and width can be configured. As in the Arria 10. but the port’s onboard memory can be configured in the manner shown in Table 2:

For the feature map memory, the total storage capacity is the product of the maximum throughput and the depth, which can be expressed as: (11)$$C_{FT}=Ceil\left(N_{in}\times {(N}_{y}+K_{Max}\;-1)\times D_{f}\right)\;Bytes$$
where $K_{Max}$ is the maximum convolution kernel width that can be executed.

Whereas in the purposed design, we set the maximum number of non-zero values allowed for the number of non-zero values within each convolutional kernel to be $D_{w}$, so that the memory required for the
(12)$$C_{WT}=D_{w}\times N_{scu}\;Byte$$

We need another memory to store the locations of the non-zero values, and the locations of the nonzero weights are represented by the tour codes as n bits, the memory required to store non-zero valued locations is: (13)$$C_{pos}=D_{w}\times \frac{n}{8}\times N_{scu}\;Bytes$$

**Global Memory Port Configuration.** In summary, the total utilization of RAM resources is shown in Equation (Equation 14): (14)$$C_{RAM}=C_{FT}+C_{pos}+C_{WT}+C_{0}$$
where $C_{0}$ is otherwise configurable as RAM at the time of optimization Block uses the resources consumed by the logic.

**DSP Resources.** In designing the multiply-accumulate computation unit, one DSP supports two 8-bit multiplication operations, and the number of DSPs consumed in the multiply-accumulate kernel used in the solid computation unit $C_{DSP,CONV}$ can be obtained from Equation (Equation 15):(15)$$C_{DSP,CONV}=\;\frac{{(N}_{scu}\times N_{in}\times N_{y})}{2}$$

The total DSP usage C_DSP_ is shown in Equation (Equation 16):(16)$$C_{DSP}=C_{DSP,CONV}+C1\times N_{scu}+C_{2}$$
where $C_{1}$ is the DSP resources occupied in the computation unit for use in other multiplication calculations, and $C_{2}$ is the DSP resources consumed outside of the computation unit (including the scheduler, DSU (memRead) and LSU (memWrite)).

**Logic Element (LE).** The use of logic resources for the architecture as a whole is shown in Equation (Equation 17), where $C_{3}$ is the constant within the compute unit that does not grow with parallelism, $C_{4}$ denotes the constant within the compute unit that grows with parallelism, and $C_{5}$ denotes the constant that is intrinsic to the compute unit outside the compute unit.
(17)$$C_{RAM}=\left(C_{3}+C_{4}\times N_{in}\right)\times N_{y}\times N_{scu}+C_{5}$$

In subsequent experiments, we use this model to establish theoretical benchmarks, evaluate the actual performance of the design, and identify bottlenecks by comparing the experimentally measured values against the theoretical predictions.

### 4.3. Design Space Exploration (DSE)

In the final design, we choose the Algorithm 1 for the experiments. We build cycle-accurate simulator for emulating the actual performance. The effect between the number of packets accessed by the scheduler and the conflicts and pauses is shown in Figure 17a,b. The results are summarized as follows.

On this basis, we summarize the results of the design space exploration, shown in Table 3, where we explore the $N_{in}$, $N_{y}$, $N_{scu}$ directions respectively and set the corresponding reasonable BANK_NUM for the number of parallel computational units to ensure the throughput rate of the feature map memory. The following table shows the experimental results of design space exploration.

Then, we automate the compilation of the target file and generation of the report using a series of pre-defined $N_{in}$ and $N_{scu}$ parameters. The results of each design space exploration in Figure 18 appear as a point in the graph, with the execution time being the complete average execution time for each image. Hardware resources and performance generally grow in a linear fashion for the same $W_{f}$, so in the event that we are unable to complete the wiring, we estimate the overall design space utilizing RTL in the form of optimized hardware resource estimates. In our design space exploration, we observed that reducing the number of compute units (i.e., $N_{in}$) is beneficial for conserving logic resources. However, fewer compute units result in each unit handling a greater number of fan-outs, which significantly impacts the operating frequency, as illustrated in Table 3.

### 4.4. Efficiency Analysis

Additionally, due to the grouping nature and shared feature map design, a reduction in the number of computational units decreases the amount of computation performed per read. As a result, the computational density declines as the number of computational units is reduced.In the network efficiency analysis, we observed that the size of the convolutional kernel significantly affects efficiency. Convolutional kernels that are too small (e.g., 1 × 1 convolutions) exhibit reduced efficiency due to limited opportunities for data reuse. We visualize the relationship between Feature Map and Kernel in Figure 19.

We can obtain the theoretical execution time and throughput per layer by evaluating the model in Section 4.2.1 and the sparsity level after pruning. Subsequently, obtaining the execution time of each layer of the gas pedal execution in the experiments will allow us to build the Roofline Model shown in Figure 20, as well as the efficiency analysis in Figure 21.

Based on the comparison we can conclude that most of the efficiently performed convolutional layers occur when the feature graph size is small. This is because for a given number of channels, the smaller the feature map size, the smaller the number of chunks in the convolution, and the fewer the number of prefetching window slides; which means that the number of times the data is handled is smaller. On the other hand, we can also conclude that i.e., the execution efficiency of convolution kernel size of K = 3 at the same feature graph size is significantly better than that of convolution kernel of K = 1. This can be explained by the data reuse strategy we mentioned in the previous section, because the amount of data that can be reused is drastically reduced by the reduction of the convolutional kernel size, and at the same time, in order to minimize the waste of RAM resources, we try to keep the $D_{f}$ as small as possible, so the number of times the data can be reused decreases when the channel is deeper and K = 1. This brings about performance degradation. However, on devices that can satisfy network execution, we can still adjust the parameter $D_{f}$ to control the balance between resource utilization and execution efficiency.

### 4.5. Analysis and Comparison

We compared the performance and design cost of the proposed accelerator with previous works in Table 4. Notably, the performance figure of 1827 GOP/s represents the throughput of the equivalent non-sparse convolution. Equivalent throughput is a key measure of computing speed under the metric of standard network (dense) computation. The size of the equivalent is related to the sparsity rate in the compression of the neural network model. Therefore, across networks and tasks, the effect of the sparsity rate on the equivalent throughput will vary. The work in this paper achieves 48.9% throughput compared with GPU with 13.4% energy consumption.

DNNVM [51], SparkNOC [52], FlexCNN [15] and 3D-VNPU [54] used dense convolution for the operation. Because of the simpler architecture of the dense convolution accelerator design, and without more control logic to be aware of the sparsity. Therefore, the dense convolution often achieves higher actual throughput. DNNVM [51] uses ahead-of-time compilation(AoT) to hide latency during inference by schedule the order of operations in the network, which is quite similar to our purposed work. In a convolution core, multi PEs is implemented to perform multiply and the result is reduced to the accumulator. This achieves the third highest thoughput but with over 500 MHz of clock speed. The throughput is much lower than our proposed network compared with ops per clock. SparkNOC [52] uses 16-bit precision operation under dense matrix multiplication operation, and implemented with line-buffer to perform convolution. A sequence of cores including convolution core, and pooling core is connected for dataflow, which requires a lot of resource consumption for FPGA. A tree accumulating adders is also implemented for reducing latency. Besides, the target network for these is small, so the pressure to compute units and memory bandwidth is also relatively small. The experimental results also show that the energy consumption ratio and computational density are still at a disadvantage, despite the high DSP utilization of the FPGA and the high efficiency of the convolution execution. ASC’21 [53] accelerated RseNet50 using 16-bit data format, also, a special arrangement of data in memory was designed to optimize bandwidth access. The frequency of the tree formation accumulation computational unit is reduced to only 150 MHz. FlexCNN [15] achieves the highest Throughput while using the largest FPGA card with 1728k LuTs, and a faster 3D-VNPU is an accelerator based on the Winograd-based decomposition method. Also, other optimizations such as dynamic tiling, layer fusion, and data layout optimizations is also implemented to get better performance. The main architecture of this work is based on a systolic engine with transposed, and dilated convolutions supported. A multi-channel dataloader is also introduced to load featuremap and weights. It is implemented on an FPGA with 16 nm lithography technology, which engages a better power performance. However, because of the restrictions of DSPs, the throughput is limited to 1150 GB/s.

While ISARC’17 [50], VLSI’20 [4], TCAS1’21 [33], and FCCM’22 [34] use sparse convolution as the operation. Among them, ISARC’17 [50] designs the convolution kernel and fully connected kernel separately in the design, which wastes a lot of logic resources and leads to a decrease in efficiency in terms of making an equivalent transformation of the fully connected layer in this paper. Similar to our purposed work, TCAS1’22 [33] also proposed a weight reordering algorithm to preprocess compressed weights and balance the workload of PEs. An adaptive on-chip dataflow, namely hybrid parallel (HP) dataflow, is introduced to promote weight reuse. Because of the architecture design which needs to support workload balance, a crossbar is introduced during the connection of the weight bank and compute units, which burdens the logic utilization, and also the floorplan. Compared with our performance, TCAS1’22 only achieves 20.8% of throughput at 90.9GOP/S, considering the target FPGA SX660 has 57% of area at 660k LUTs, compared to our target FPGA board with 1150k LUTs. The VLSI’20 [4] accelerator focuses on structured sparse CNN, it implements a VGM to address the sparsity with shared indexes. Note that a structured pruning model requires less control logic to keep high efficiency because the data arrangement is structured and is also friendly to parallel processing as well as memory access. It achieves the actual throughput at 990 GOP/s with an equivalent throughput of 1565 GOP/s at 8-bit accuracy. Thanks to the simple design brought by the support limited to structured sparse convolution, it can achieves a higher actual throughput on a acceptable larger target FPGA chip with similar clock frequency. But because the coarse-grained structured sparsity, it does not provide a larger sparsity without any latency loss, so a lower equivalent throughput is expected. Comparing the results provided by PyTorch’s standard implementation in ResNet50 on the Nvidia RTX 2080Ti GPU, this paper uses smaller input image parameters for inference and achieves an equivalent throughput of 1827 GOPs within a loss of 1% accuracy. Additionally, the chip process and off-chip cache are key factors in the operating frequency and inference speed; the RTX 2080Ti GPU is equipped with GDDR6 memory on a 12nm lithography technology, whereas this work uses a 20nm lithography technology implemented with DDR3 memory.

## 5. Conclusions

In this paper, we present the design of accelerators capable of efficiently deploying sparse convolution operations, addressing the challenges associated with sparse convolution computation following unstructured pruning. On one hand, the accelerator architecture is carefully designed to balance computational power and memory access rates, optimizing external memory utilization. This ensures that the convolution kernel executes with maximum efficiency. Furthermore, the design incorporates zero-value detection in CNNs, effectively reducing unnecessary computations and memory overhead. This enhancement improves operational speed and reduces power consumption. On the other hand, an efficient, consistent-access register with a synchronization mechanism is designed to provide high-bandwidth chip access memory for multiple computing units. The data storage structure is co-designed with the convolution kernel to enhance the data reuse rate, reduce memory transmission, and improve the efficiency of both memory access and convolution operations. Experimental results demonstrate that the proposed hardware architecture effectively alleviates the pressure on DRAM resources within the FPGA chip. By maximizing the utilization of various logic resources and achieving a balanced design, the architecture delivers superior performance.

Future work can be explored in the following two directions: (1) Implementing lower-precision CNN quantization. Binary networks have demonstrated excellent performance in various learning tasks and are particularly well-suited for FPGA deployment. Developing accelerators for binary convolution operations represents a promising direction for future advancements. (2) We observe that when the feature map is processed through the ReLU layer, a significant number of zero values are generated, which do not contribute to subsequent computations. Leveraging this sparsity in the feature map can further reduce computational overhead and enhance equivalent computing power. For instance, as shown in the figure, the activation sparsity in convolutional neural networks like VGGNet and AlexNet can reach up to 60%. This presents an opportunity to significantly decrease the computational workload or improve performance. We reserve the exploration of this feature for future work, with feature map sparsification being a key focus of subsequent research.

## Figures and Tables

**Figure 1 micromachines-16-00022-f001:**
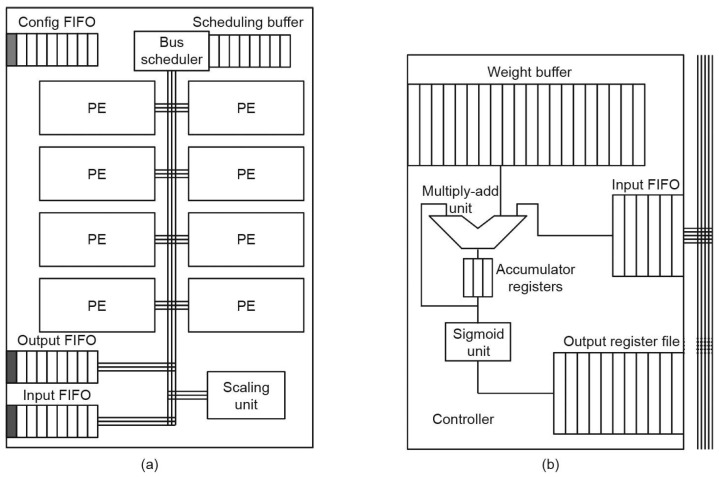
TPU architecture diagram, subfigure (**a**) illustrates the overall TPU architecture design. (**b**) illustrates the structure of the compute unit in each PE. [21].

**Figure 2 micromachines-16-00022-f002:**
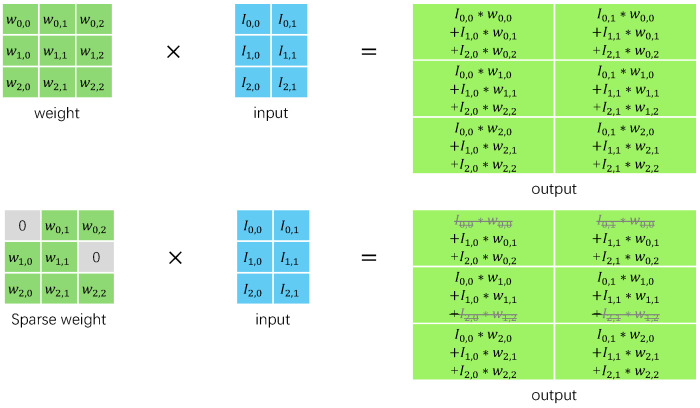
Illustration of how sparse convolution is conducted.

**Figure 3 micromachines-16-00022-f003:**
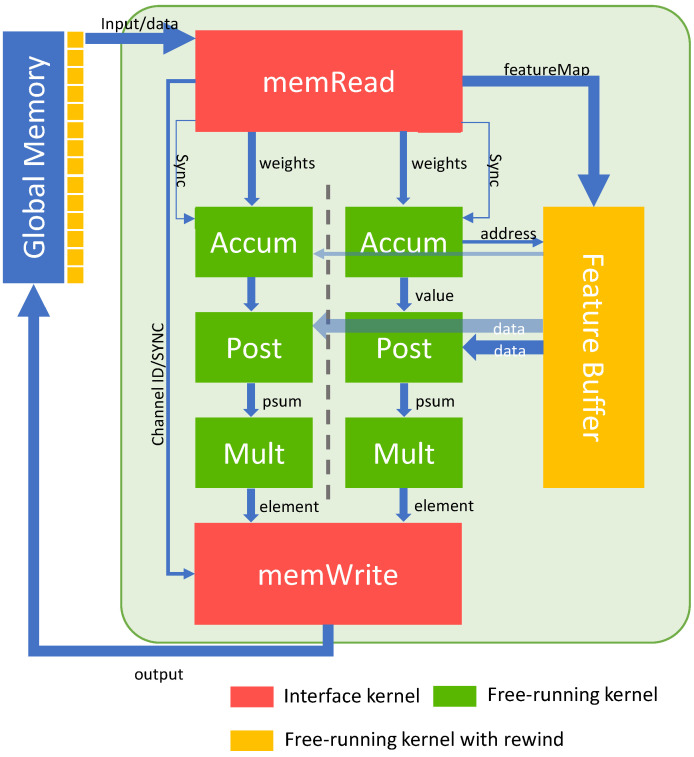
The overall architecture of proposed accelerator.

**Figure 4 micromachines-16-00022-f004:**
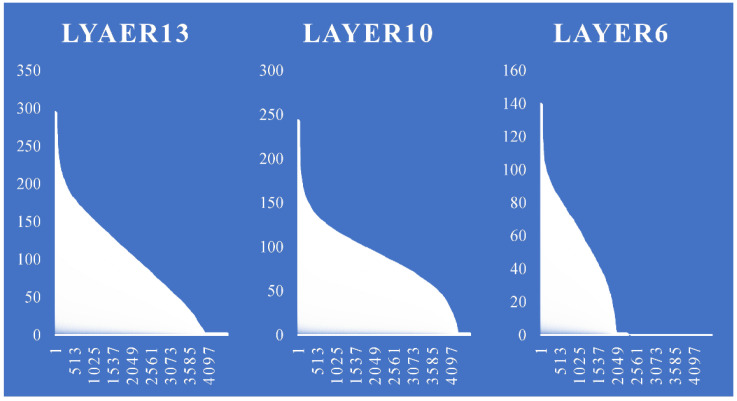
Channel non-zero number(workload) in VGG16.

**Figure 5 micromachines-16-00022-f005:**
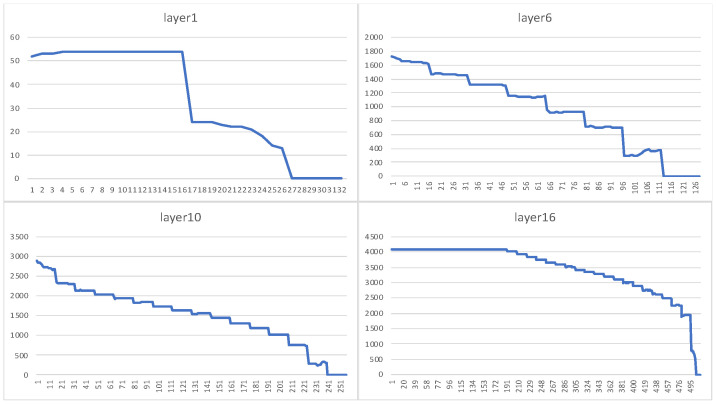
Channel work balance over PE.

**Figure 6 micromachines-16-00022-f006:**
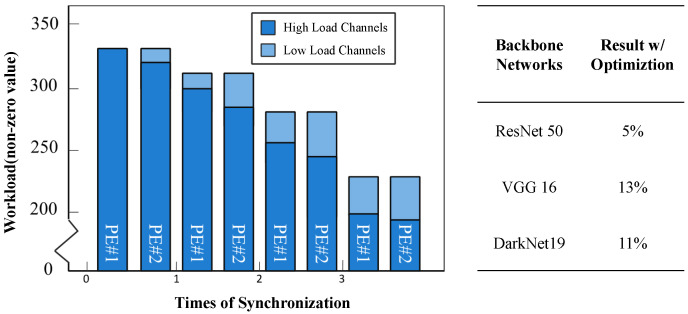
Bank execute order intra PE.

**Figure 7 micromachines-16-00022-f007:**
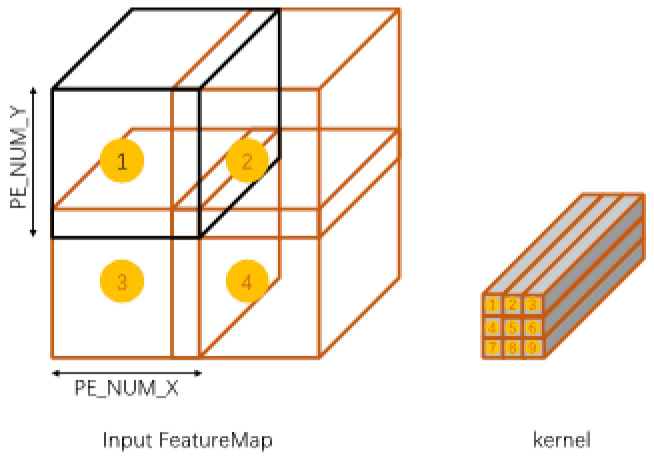
Prefetch window parallelism scheme.

**Figure 8 micromachines-16-00022-f008:**
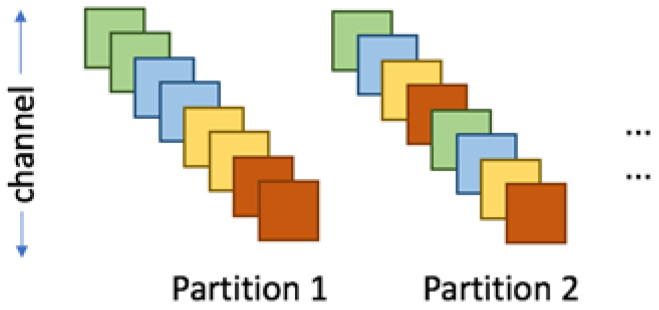
Intra channel array partitioning scheme.

**Figure 9 micromachines-16-00022-f009:**
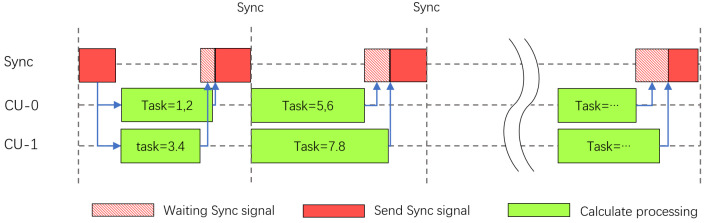
Synchronization scheme of parallel convolution tasks.

**Figure 10 micromachines-16-00022-f010:**
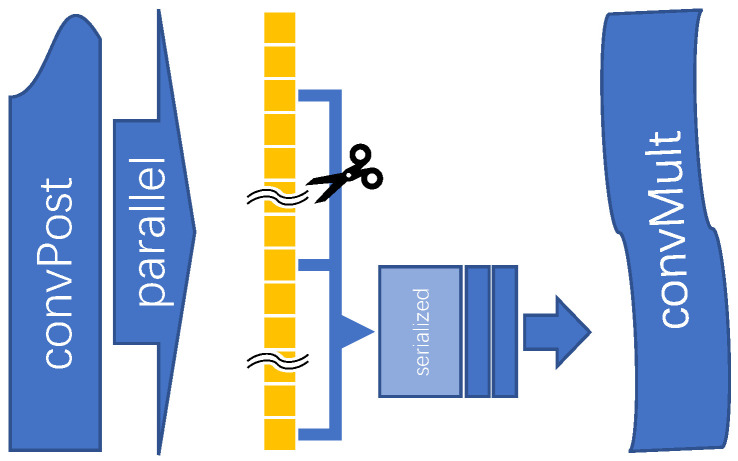
Serialization of partial sum based on streaming.

**Figure 11 micromachines-16-00022-f011:**
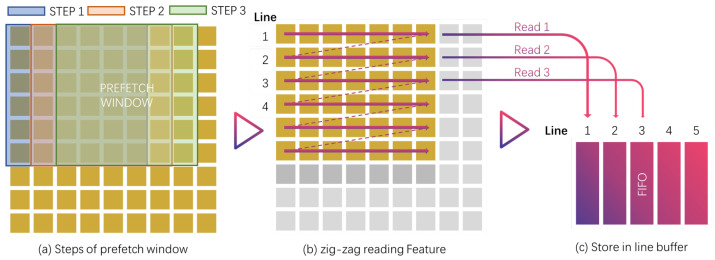
Sliding-window based fetching approach of Feature Map data.

**Figure 12 micromachines-16-00022-f012:**
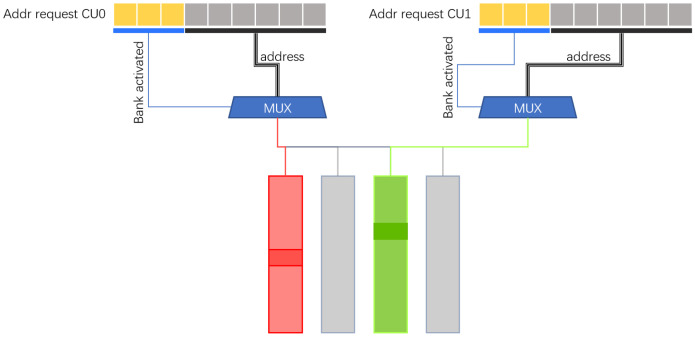
Multiple bank R/W design.

**Figure 13 micromachines-16-00022-f013:**
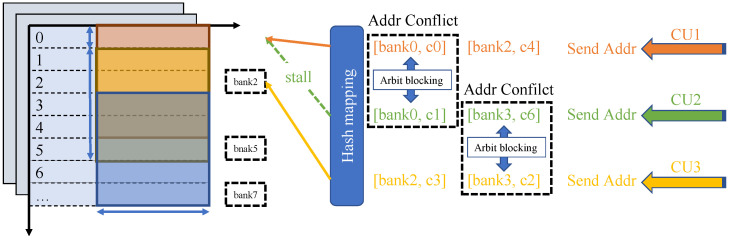
Overall architecture of hash shared memory execution diagram.

**Figure 14 micromachines-16-00022-f014:**
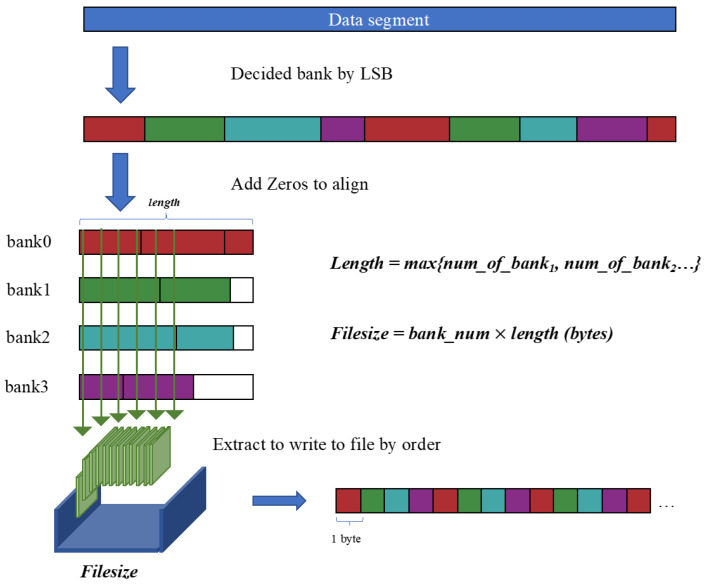
Data structure of stored weight file.

**Figure 15 micromachines-16-00022-f015:**
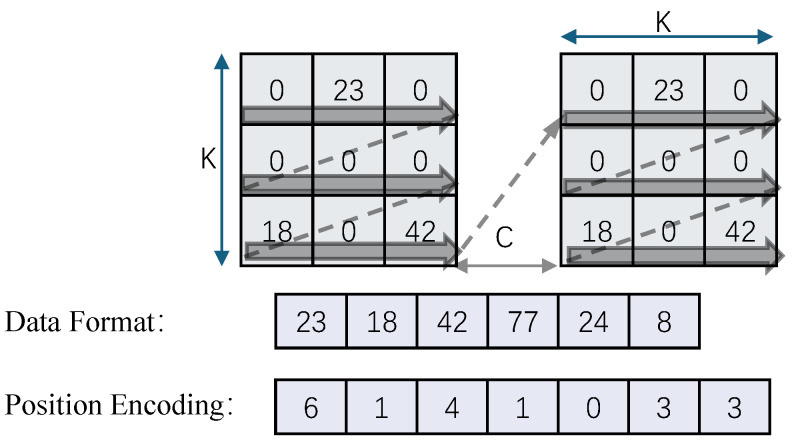
Weight encoding scheme.

**Figure 16 micromachines-16-00022-f016:**
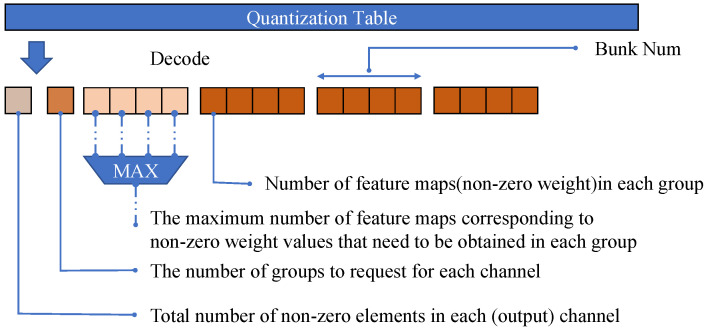
Data format of quantization table.

**Figure 17 micromachines-16-00022-f017:**
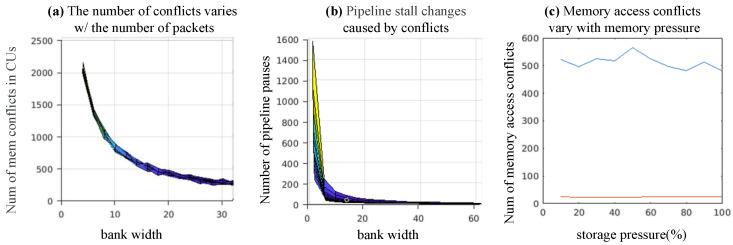
DSE of the shared memory bank selection.

**Figure 18 micromachines-16-00022-f018:**
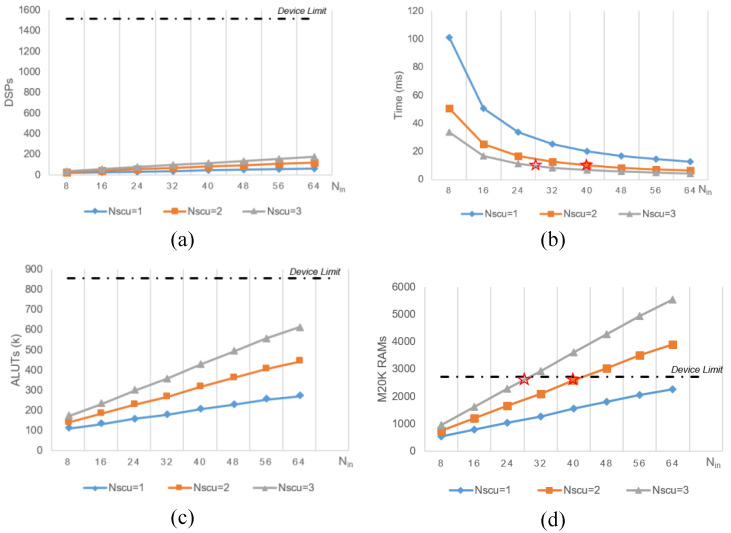
DSE of the design parameters. (**a**) shows the DSPs usage over $N_{in}$ after floorplanning. And the roofline limited by device is marked as dotted line. (**b**) shows the wall time of network reference over $N_{in}$. The optimum design is marked in a red star. (**c**) shows the ALUTs usage with the increase of the parallel. and (**d**) shows the on-chip RAMs usage over $N_{in}$, and the optimum design is found by the elimination of device resource, marked in red star.

**Figure 19 micromachines-16-00022-f019:**
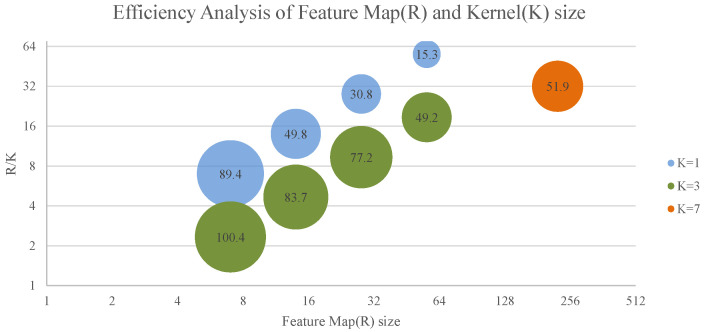
Efficiency of different kernel and Feature Map size.

**Figure 20 micromachines-16-00022-f020:**
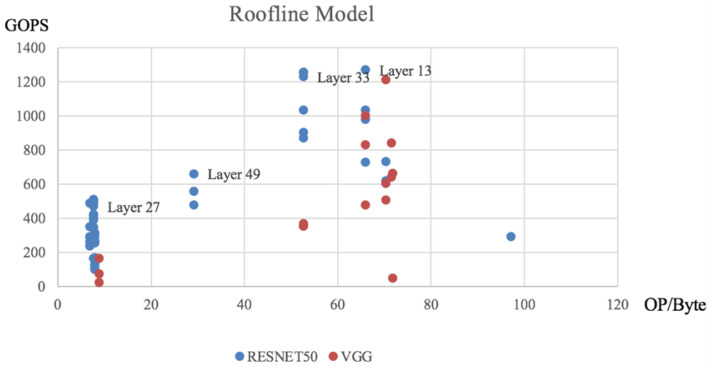
Comparision of the measured time, theoretic time, and efficiency.

**Figure 21 micromachines-16-00022-f021:**
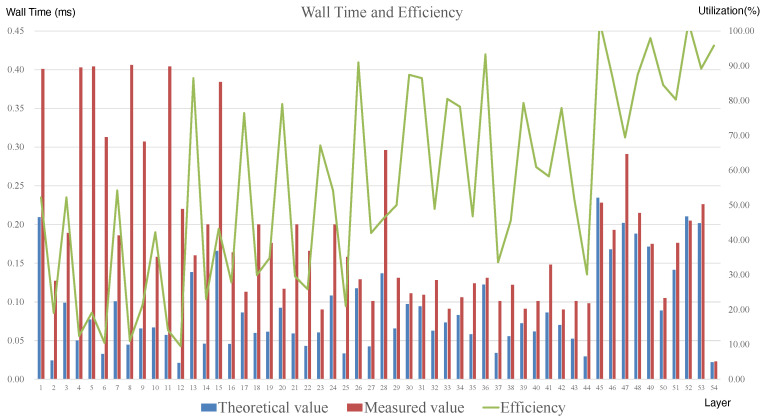
Roofline Model data point.

**Table 1 micromachines-16-00022-t001:** Hash algorithm performance summary.

Hash Algo.	Conflict Rate	Utilization	Vacancy Rate	Remapping
XOR-based Hashing [40]	0.117188	0.550781	0.332031	132
linear probing [41]	0.017578	0.761719	0.220703	116
one-at-a-time(Jenkins) hash [42]	0.189453	0.775391	0.035156	-
perfect hash [43]	0.181641	0.400391	0.417969	-
mid-square [44]	0.173828	0.242188	0.583984	-
folding mathod [46]	0.035156	0.962891	0.001953	-
modulo arithmetic [47]	0.1875	0.402344	0.410156	-
revert LSB [45]	0.111248	1.000000	0.000000	-

**Table 2 micromachines-16-00022-t002:** Arria GX1150 SRAM configuration.

Memory Block	Depth (Bits)	Programmable Width
M20K	512	×40, ×32
M20K	1K	×20, ×16
M20K	2K	×10, ×8
M20K	4K	×5, ×4
M20K	8K	×2
M20K	16K	×1

**Table 3 micromachines-16-00022-t003:** Summary of the design space exploration results.

	ResNet50	VGG16	ResNet50	VGG16
network infrastructure	4 BANK	4 BANK	16 BANK	16 BANK
$N_{in}$	14	14	14	28
$N_{y}$	20	14	20	20
$N_{scu}$	4	4	4	2
Bank Number	16	16	4	4
frequency (MHz)	222.22	195.65	211.18	180.05
throughput rate (GOPS)	497.77	437.91	472.64	403.2
equivalent numerical power (GOPS)	1579.59	1827.88	1499.83	1683.36

**Table 4 micromachines-16-00022-t004:** Comparison with other accelerator designs.

	Platform	Tools	Backbone	Inputs	Qua. ***	Sparse	Power (W)	Energy Efficiency (GOP/W)	Top1 (%)	Throughput (GOP/S)
ISARC’17 [50]	ZC706 **	Vivado	ResNet50	-	32	-	1.167	0.97	-	-
DNNVM [51]	ZU9 **	Vivado	VGG	224 × 224	32	N	22.8	123	69.6	1780
SparkNOC [52]	GX1150 *	OpenCL	SparkNet	800 × 600	16	N	21	0.04	73.54%	337.2
AES’21 [53]	ZCU102 **	Vivado	ResNet50	224 × 224	16	Y	23.6	-	-	309
FlexCNN [15]	U250 **	Vitis	VGG16	224 × 224	8	N	-	-	82.1%	2329.1
3D-VNPU [54]	ZCU102	Vitis	VGG16	224 × 224	8	N	10.2	121	-	1150
TCAS1’22 [33]	SX660 *	Verilog	ResNet50	224 × 224	16	Y	4.6	23.42	-	90.9
VLSI’21 [4]	ZCU102	Vivado	VGG16	-	8	Y	17.1	19.61	-	1565
VLSI’21 [4]	ZCU102	Vivado	VGG16	-	16	Y	15.4	18.66	-	782
Ours	GX1150 *	OpenCL	ResNet50	224 × 224	8	Y	26	70	76.17%	1827

* represent a product of Altera, an Intel company. ** represent a product of Xilinx. *** quantization bitwidth.

## Data Availability

No new data were created or analyzed in this study. Data sharing is not applicable to this article.

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
