# Peer review of "Sparse Convolution FPGA Accelerator Based on Multi-Bank Hash Selection"

_micromachines, 2024, doi:10.3390/mi16010022_

Round 1

Reviewer 1 Report

Comments and Suggestions for Authors

The work proposes a hardware accelerator for sparse convolutional operations for neural network workloads. In detail, the proposed accelerator includes a mechanism to improve the off-chip access bandwidth utilization through a shared feature map cache memory using hash approaches.

The general organization of the manuscripts is clear. However, several adjustments are required in the document to better highlight and justify the main contributions.

The reviewer suggests adding as keywords: cache memory

To highlight the main outcomes of the work, the reviewer strongly suggests adding the main contributions of the work as a bullet list in section 1 (Introduction).

Moreover, motivation and research problems can be improved in the Introduction section. In the current state, it is unclear which of the indicated issues are faced by the current work.

Lanes 52 and 53 seem not well linked with the paragraph and require further improvement.

Section 2 (Related works) requires improvements. Several relevant hardware accelerator architectures for DNNs are not listed/cited in the manuscript (e.g., LPUs, Tesla autopilot HW).

In particular, the target of Section 2.1 is not clear. Is it focused on the hardware? or is describing the algorithms required to deploy DNN workloads (e.g., convolutions as matrix multiplications or FFT)?. Moreover, it is incomplete. There is missing information regarding the vector extensions on processors and the internal hardware accelerator extensions on modern CPUs and GPUs for DNN workloads (e.g., Tensor/matrix cores).

The reviewer strongly recommends clarifying section 2. Moreover, the last part of the paragraph (lines 125 to 131) is unclear. Are the authors highlighting the approach employed by their design or comparing it with other works in the state of the art?

Section 2.2 seems to focus on soft accelerators for FPGAs, which must be named accordingly.

To simplify reading, Section 3 must be described in present tense (simple present, e.g., This chapter analyses …)

The description of the structures in subsection 3.1 is clear. However, the main motivations for the selection of the hardware structures are unclear at all. In this subsection, the algorithm implemented and used during the execution of the hardware accelerator must be transparent. However, in the current state, it is not discussed or explained to justify the selection of the structures.

The design description (Section 3) of the method or approach used to implement sparsity to the inputs is unclear.

What is the sparsity technique used?, Please elaborate and clarify.

Figure 20 is missing the x-axis and y-axis legends.

The conclusion focuses on future work instead of highlighting the current results obtained. Please update and clarify.

Comments on the Quality of English Language

The text is acceptable. However, to improve the legibility of the manuscript for any reader, the reviewer strongly suggests verifying the text, if possible by a native speaker.

Reviewer 2 Report

Comments and Suggestions for Authors

The paper conducts in-depth research on the design of an FPGA-based sparse convolution computation acceleration circuit, proposing a novel neural network accelerator design scheme and validating its effectiveness through experiments. The paper demonstrates a certain degree of innovation in its technical approach, including strategies such as utilizing compiler optimization plugins to automatically optimize address power and designing hashing methods to avoid conflicts. Additionally, the experimental section of the paper is relatively comprehensive, effectively supporting its conclusions.

However, the paper also has some shortcomings. For example, when introducing related work, it lacks sufficient discussion and comparison of the latest research progress and cutting-edge technologies in the field of sparse neural network acceleration. Furthermore, although the experimental section presents comparison results with other related work, the analysis and interpretation of the experimental data are not deep and comprehensive enough.   It can be accepted after making the aforementioned revisions.
